# The Utility of Contrast-Enhanced Ultrasound (CEUS) in Assessing the Risk of Malignancy in Thyroid Nodules

**DOI:** 10.3390/cancers16101911

**Published:** 2024-05-17

**Authors:** Agnieszka Żyłka, Katarzyna Dobruch-Sobczak, Hanna Piotrzkowska-Wróblewska, Maciej Jędrzejczyk, Elwira Bakuła-Zalewska, Piotr Góralski, Jacek Gałczyński, Marek Dedecjus

**Affiliations:** 1Department of Endocrine Oncology and Nuclear Medicine, Maria Sklodowska-Curie National Research Institute of Oncology, 02-781 Warsaw, Poland; goralp83@gmail.com (P.G.); jacek.galczynski@pib-nio.pl (J.G.); marek.dedecjus@gmail.com (M.D.); 2Radiology Department II, Maria Sklodowska-Curie National Research Institute of Oncology, 02-034 Warsaw, Poland; kdsobczak@gmail.com; 3Department of Ultrasound, Institute of Fundamental Technological Research, Polish Academy of Sciences, 02-106 Warsaw, Poland; hpiotrzkowska@gmail.com; 4Department of Ultrasound and Mammography Diagnostics, Mazovian Brodnowski Hospital, 03-242 Warsaw, Poland; mjedrzejczyk@interia.pl; 5Department of Pathology, Maria Sklodowska-Curie National Research Institute of Oncology, 02-781 Warsaw, Poland; elwirabz@onet.eu

**Keywords:** thyroid cancer, cancer screening, clinical trial, contrast-enhanced ultrasound, thyroid lesion

## Abstract

**Simple Summary:**

Ultrasonography is a basic tool used in the evaluation of thyroid nodules, but there is no single feature of this method which predicts malignancy with statistical significance. The aim of the study is to assess the usefulness of contrast enhanced-ultrasound (CEUS) in the differential diagnosis of thyroid nodules. The highest value of the study results from the multiparameter approach to the evaluation of thyroid lesions in the light of new diagnostics methods and assessment of the unique combinations of both B-mode and CEUS features as predictors of thyroid cancers. Moreover, several qualitative contrast features predicting benign lesions were evaluated. The preliminary results indicate that CEUS is a useful tool in assessing the risk of malignancy of thyroid lesions. The combination of the qualitative enhancement parameters and B-mode sonographic features significantly increases the method’s usefulness. Further studies should be performed to introduce CEUS patterns in the diagnostic algorithm of thyroid nodules.

**Abstract:**

Background: Ultrasonography is a primary method used in the evaluation of thyroid nodules, but no single feature of this method predicts malignancy with high accuracy. Therefore, this paper aims to assess the utility of contrast-enhanced ultrasound (CEUS) in the differential diagnosis of thyroid nodules. Methods: The study group comprised 188 adult patients (155 women and 33 men) who preoperatively underwent CEUS of a thyroid nodule classified as Bethesda categories II–VI after fine-needle aspiration biopsy. During the CEUS examination, 1.5 mL of SonoVue contrast was injected intravenously, after which 15 qualitative CEUS enhancement patterns were analysed. Results: The histopathologic results comprised 65 benign thyroid nodules and 123 thyroid carcinomas. The dominant malignant CEUS features, such as hypo- and heterogeneous enhancement and slow wash-in phase, were evaluated, whereas high enhancement, ring enhancement, and a slow wash-out phase were assessed as predictors of benign lesions. Two significant combinations of B-mode and CEUS patterns were noted, namely, hypoechogenicity with heterogeneous enhancement and non-smooth margins with hypo- or iso-enhancement. Conclusions: The preliminary results indicate that CEUS is a useful tool in assessing the risk of malignancy of thyroid lesions. The combination of the qualitative enhancement parameters and B-mode sonographic features significantly increases the method’s usefulness.

## 1. Introduction

Ultrasound (US) examination of the thyroid gland is an essential tool in the diagnosis of thyroid disorders. The use of this method has revealed focal lesions in approximately 50% of the population, with malignant lesions affecting 5–15% of thyroid nodules [1,2,3]. The primary aim of US diagnosis of thyroid lesions is to estimate the risk of thyroid cancer and select lesions for fine-needle biopsy, which is generally associated with a favourable prognosis and, in more than 90% of cases, a long overall survival of 10 years [4].

The main limitation of ultrasonography is the lack of a single sonographic feature that allows the differentiation of benign and malignant focal lesions with sufficient sensitivity and specificity for the detection of thyroid cancer [5]. Over the past decade or so, a number of papers have been published elaborating on the US features of high-risk malignancy, which include the presence of microcalcifications, solid composition, hypoechogenicity, irregular margins, and a ‘taller-than-wide’ shape [6,7]. In the absence of a single sonographic pattern characteristic of a malignant thyroid lesion, classifications based on algorithms allowing the US risk stratification of cancer have emerged over the last decade [8]. In 2009, Horvath et al. developed the TIRADS (Thyroid Imaging Reporting and Data System) based on US features, proposing the classification of focal lesions into TIRADS classes 1–6, according to the increasing risk of cancer [9]. Over the years, the world’s scientific societies have introduced their own modifications to the TIRADS system, including the classifications proposed by the American Thyroid Association (ATA), the European Thyroid Association (EU-TIRADS), the American College of Radiologists (ACR-TIRADS), and the Korean Society of Thyroid Radiology (K-TIRADS), and their clinical utility and comparative evaluation have been discussed in many studies [8,10].

Diagnostic difficulties mostly concern lesions of intermediate risk of malignancy both ultrasonographically (TIRADS 3 and 4) and cytopathologically (Bethesda categories III and IV), as the decision regarding radical treatment is often made on an individual basis, based on coexisting additional clinical factors, in the absence of unequivocal cancer risk features in the lesion [11,12,13]. New diagnostic tools are therefore being sought to optimise the interpretation of US images, such as strain elastography (SE), shear wave elastography (SWE), and contrast-enhanced US (CEUS), and the use of these methods in thyroid cancer risk stratification is constantly assessed [14,15]. Nowadays, machine learning models are used in order to improve sensitivity, specificity and accuracy of diagnostic modalities, including the reliability of elastography on the classification of thyroid nodules [16].

CEUS allows the real-time imaging of vascular perfusion in the focal lesion under study, following the intravenous administration of a contrast agent composed of microvesicles containing sulphur hexafluoride surrounded by phospholipids and palmitic acid [17]. The main advantage of the CEUS method is that it causes a lower incidence of adverse reactions after contrast administration (according to estimates, 1:10,000) compared with the contrast agents used in computed tomography (CT) or magnetic resonance imaging (MRI) [18]. The CEUS method is in clinical use in different medical fields, and is involved in imaging the liver lesions and a wide range of non-hepatic indications referred to as musculoskeletal medicine or the assessment of urinary tract pathology [19,20].

CEUS allows the real-time analysis of qualitative features as well as the evaluation of quantitative parameters by determining contrast enhancement curves using special software [21]. The authors of the 2017 European Federation of Societies for Ultrasound in Medicine and Biology (EFSUMB) guidelines indicate that CEUS examination can be used in the differential diagnosis of focal thyroid lesions, and distinguish two features that are strong predictors of a malignant lesion, namely, hypoenhancement (sensitivity: 82.0%; specificity: 85.0%; accuracy: 84.0%) and heterogenous enhancement (sensitivity: 88.2%; specificity: 92.5%; accuracy: 90.4%), and the one feature of a benign lesion, namely, ring enhancement (sensitivity: 83.0%; specificity: 94.1%; accuracy: 88.5%) [22]. The CEUS method is not recommended for use in routine clinical practice, despite having been the subject of many studies, including meta-analyses, which indicate its high sensitivity and specificity in the prediction of thyroid cancer [23].

The main objective of this paper is to evaluate the clinical utility of the application of CEUS’s examination features in combination with the analysis of B-mode features, including elastography, in the differential diagnosis of focal thyroid lesions.

## 2. Material and Methods

One focal lesion was assessed in each patient. All patients were treated surgically at the Department of Oncological Endocrinology and Nuclear Medicine of the National Institute of Oncology in Warsaw. The study group was divided, depending on the histopathological examination results. Approval was received from the local ethics committee (no. 83/2021).

The inclusion criteria were as follows: adult patients over 18 years of age with a focal lesion of the thyroid gland who qualified for surgical treatment with a cytopathological diagnosis in Bethesda categories III–VI or patients in Bethesda category II with clinical symptoms of difficulties in breathing and swallowing who had given their written consent to undergo a CEUS examination.

The exclusion criteria were as follows: patients under 18 years of age, pregnant and breastfeeding women, lack of written consent to undergo a CEUS examination, and contraindications for the administration of the Sonovue contrast agent, such as severe heart muscle diseases including severe congenital or acquired heart defects, advanced heart failure, unstable ischaemic heart disease, severe cardiac arrhythmias, severe pulmonary hypertension, untreated or uncontrolled hypertension, and respiratory distress syndrome.

### 2.1. B-Mode, Elastography Examination Technique

All 188 patients underwent US examination on a high-performance US device (Philips, EpiQ 5, Bothell, DC, USA) using a linear probe eL 18-4, with 22-2 frequency range.

Images were saved in DICOM and on memory loops (2.5 min videos were recorded). Three measurements of the gland and focal lesions were made: length (in longitudinal position of a transducer), depth, and width (in transverse position of a transducer).

The volume of the gland was calculated based on three dimensions. The morphological features of focal lesions in US examination were assessed according to the risk scale based on the EU-TIRADS classification, including composition (solid, mixed predominantly solid, mixed predominantly cystic, spongiform, or cystic), shape (oval, round, irregular, taller-than-wide, taller-than-long), margin (smooth, irregular, ill-defined), macro and microcalcification, echogenicity (isoechoic, hyperechoic, mildly hypoechoic, markedly hypoechoic), echotexture (heterogenous, homogenous), and type of vascularity.

The 4-grade Asteria scale was used for SE assessment. The cut-off values for deformable (soft) lesions and non-deformable lesions were Asteria 1 and 2 and Asteria 3 and 4, respectively. While performing the elastography examinations, the endocrinologist was careful to avoid compressing the neck with the probe to minimise false-positive results.

US examinations were performed by an experienced endocrinologist, after which the saved images were subjected to B-mode US assessment by two clinicians with at least 10 years of experience in the assessment of thyroid tumours. Firstly, all US images and DICOM memory loops were evaluated by specialists, separately. In the second step, all databases were discussed together and interobserver disagreements were assessed in approximately ten per cent of cases, in which consensus was reached decisively.

### 2.2. CEUS Examination Technique

After a B-mode US examination was performed and the focal lesion of interest was determined in each patient, a CEUS examination was performed using the US device (Philips, EpiQ 5) with a linear probe with a frequency of 18.0 MHz. In all patients in the study group, 1.5 mL of contrast agent (Sonovue, Bracco; Milan, Italy) was administered into a peripheral vein immediately after collection into a syringe, and after each injection, 5 mL of 0.9% sodium chloride solution for injection was administered through the same intravenous access. Thereafter, a CEUS examination of the selected focal lesion was performed during a 2.5 min recording, which was then archived in DICOM files. CEUS was performed by a single investigator, and assessment of contrast-enhancement qualitative patterns was performed retrospectively by two investigators with more than 10 years of experience in US. In total, 15 contrast-enhancement parameters were assessed to compare the nature of the enhancement with the surrounding thyroid parenchyma: intensity (hypoenhancement, high enhancement, equal to thyroid parenchyma), uniformity (heterogenous, homogenous), tendency (centripetal, centrifugal, diffuse), wash-in (fast, slow, equal to thyroid parenchyma), wash-out (fast, slow, equal to thyroid parenchyma), and ring enhancement (Figure 1).

### 2.3. Pathology Examination

All patients first underwent fine-needle aspiration biopsy of a focal lesion in the thyroid gland selected for biopsy based on a US examination performed by an experienced sonographer (radiologist or endocrinologist). For each focal lesion examined, a cytopathological result was obtained according to the Bethesda system, which constitutes the basis for qualifying the patient for radical treatment.

All lesions designated for cytological examination were verified in a histopathological examination after surgical treatment.

### 2.4. Statistical Analysis

Data were analysed in software R: A Language and Environment for Statistical Computing, version 4.1.2. Numeric variables were summarised with mean and standard deviation or median and interquartile range (IQR), depending on normality of distribution. Nominal variables were presented with absolute numbers of observations and group percentages. Normality was validated with the Shapiro–Wilk test, as well as skewness and kurtosis. Variance homogeneity was checked with the Levene test. Malignant and benign groups were compared with the t-Student test, Mann–Whitney U test, Pearson Chi-square test, or Fisher exact test, as appropriate. Logistic regression was performed in two steps, namely, univariate and multivariate models. Univariate logistic regression models were run to understand which predictors were correlated with malignant nodules. The *p*-value cut-off for the selection of variables for the second step was 0.25, after which a stepwise procedure was employed to indicate the final predictors. The association of significant predictors with the odds of a malignant outcome was expressed using odds ratios (ORs) and 95% confidence intervals (CIs). All statistical tests assumed significance when the *p*-value was lower than 0.05.

## 3. Results

### 3.1. Patients

A total of 188 patients (155 women and 33 men) diagnosed with a focal lesion in the thyroid gland were included in the study. Postoperative histopathological analysis revealed a diagnosis of thyroid cancer in 123 focal lesions (100 women, 23 men, mean age 46.5 ± 13.9) and 65 benign lesions (55 women, 10 men, mean age 49.8 ± 13.5). The cancerous lesions were operated on by total thyroidectomy or removal of one lobe of the thyroid gland with isthmus and lymphadenectomy of the central compartment of the neck. The final qualification relating to the scope of the procedure was carried out by a surgeon experienced in the field of oncological surgery after analysis of the US examination, the cytopathological report, and the patient’s clinical symptoms.

The median dimensions were 13.00 mm (9.00;20.75, 95% CI) for malignant lesions and 21.5 mm (15.00;28.00, 95% CI) for benign lesions. Benign lesions were statistically significantly larger than malignant neoplastic lesions (*p* < 0.001).

### 3.2. B-Mode Findings

In the group of malignant neoplastic lesions, the EU-TIRADS 5 category predominated (*n* = 106/123, 86.2%), and there were significantly fewer tumours in the intermediate and low-risk EU-TIRADS categories 4 and 3 (*n* = 11 [8.9%], vs. *n* = 6 [4.9%], respectively). In the group of benign lesions, a fairly even distribution of focal lesions of individual EU-TIRADS categories was found, amounting to EU-TIRADS 5 *n* = 23 (35.4%), EU-TIRADS 4 *n* = 22 (33.8%), and EU-TIRADS 3 *n* = 20 (30.8%). In malignant lesions, the following features were statistically significantly more common: solid composition, irregular shape, irregular or ill-defined margins, the presence of microcalcifications, and a degree of deformability in SE, assessed as Asteria score 4 (Table 1). Moreover, the data presented in Table 1 show the features that differentiated relevantly the nature of the focal lesions. The whole statistical analysis is given in the Appendix A [Table A1, Table A2 and Table A3].

Univariate logistic regression models revealed that size significantly impacted the odds of cancer: an additional 1 mm would decrease the odds by 4% (OR = 0.96 CI_95_ [0.93;0.98], *p* = 0.002). Compared to mild hypoechogenicity, marked hypoechogenicity was associated with three-times-higher odds of cancer (OR = 3.41 CI_95_ [1.59;7.76], *p* = 0.002). A combination of mild and marked hypoechogenicity was associated with 13-times-higher odds of a malignant nodule compared to hyperechogenic or isoechogenic lesions (OR = 13.00 CI_95_ [5.29;37.05], *p* < 0.001). Moreover, margin type had a significant impact on the risk of thyroid cancer, with odds increasing 9- and 13-fold with ill-defined or irregular margins (OR = 9.05 CI_95_ [4.12;21.08], *p* < 0.001 and OR = 13.08 CI_95_ [5.58;33.50], *p* < 0.001), respectively. Furthermore, microcalcifications were associated with 10-times-higher odds of cancer (OR = 9.96 CI_95_ [3.41;42.48], *p* < 0.001). In SE, an Asteria score of 4 was related to a seven-times-higher risk of malignant thyroid lesions (OR = 7.34 CI_95_ [1.63;35.92], *p* = 0.010) (Table 2).

A multivariate logistic regression model was built of parameters from the B-mode section, CEUS section, and combinations of the B-mode and CEUS. The full outcome of the multivariate model can be read jointly from Table 2, Table 3 and Table 4 and from Table A2 in the Appendix A.

### 3.3. CEUS Findings

All patients qualified for CEUS examination. Qualitative features of contrast enhancement evaluated in the study differed significantly between malignant and benign groups. Dominant features were evaluated and compared in the groups with malignant lesions and benign tumours as follows: hypoenhancement (*n* = 44 [35.8] vs. *n* = 11 [16.9%], *p* < 0.001), heterogeneous enhancement (*n* = 101 [82.1%] vs. 40 [61.5%], *p* = 0.003), and a slow wash-in phase (*n* = 40 [32.5%] vs. *n* = 10 [15.4%], *p* = 0.018); Figure 2.

It was found that some features were more dominant in the group with benign lesions than that with thyroid cancers: high enhancement (*n* = 36 [55.4%] vs. *n* = 33 [26.8%], *p* < 0.002), a fast wash-in phase (*n* = 26 [40.0%] vs. *n* = 29 [23.6%], *p* = 0.029), ring enhancement (*n* = 11 [16.9%) vs. *n* = 5 [4.1%], and a slow wash-out phase (*n* = 19 [29.2%] vs. *n* = 12 [9.8%]); Table 5, Figure 3.

Secondly, univariate logistic regression models were performed for qualitative contrast enhancement features in the CEUS examination. One feature was identified that significantly increased the risk of thyroid cancer in the lesion, namely, heterogeneous enhancement (OR = 2.87, CI_95_ [3.41;42.48], *p* = 0.02), while for the prediction of benign lesions, five features were identified that significantly reduced the risk of malignant change, namely, high enhancement (OR = 0.23, CI_95_ [0.10;0.50], *p* < 0.001), ring enhancement (OR = 0.21, CI_95_ [0.06;0.60], *p* = 0.005), and wash-out (slow, fast, or equal to the thyroid parenchyma) compared to no wash-out (OR = 0.05, CI_95_ [0.01;0.22], *p* < 0.001 vs. OR = 0.16, CI_95_ [0.02;0.60], *p* = 0.018 vs. OR = 0.17 CI_95_ [0.03;0.66], *p* = 0.025), respectively (Table 5).

### 3.4. The Combination of B-Mode and CEUS Findings

In the univariate regression model, favourable combinations of B-mode features and qualitative features of contrast enhancement in CEUS were noted that significantly increased the risk of cancer in the focal lesion.

Echogenicity of the focal lesion assessed as hypoechogenicity (mild or marked hypoechogenicity) combined with hypoenhancement or equal-to-thyroid parenchyma enhancement resulted in eight-times-higher odds of malignant nodules (OR = 7.75, CI_95_ [3.95;15.93], *p* < 0.001).

Moreover, it was found that the hypoechogenicity of the focal lesion (mild or marked hypoechogenicity) analysed with heterogeneous contrast enhancement increased the odds of malignancy of the focal lesion by almost seven times (OR = 6.6, CI_95_ [3.42;13.10], *p* < 0.001).

Furthermore, it was shown that marked hypoechogenicity combined with hypoenhancement or equal-to-thyroid parenchyma enhancement or combined with heterogenous enhancement significantly influenced the prediction of thyroid cancer, increasing the odds 22- and 9-fold, respectively (OR = 21.58, CI_95_ [6.34;135.23], *p* < 0.001 vs. OR = 8.77, CI_95_ [3.78;24.07], *p* < 0.001, respectively).

Then, it was observed that the presence of non-smooth margins (irregular or ill-defined) with hypoenhancement or equal-to-thyroid parenchyma enhancement or with heterogenous enhancement increased the odds of malignancy ninefold (OR = 9.47, CI_95_ [4.63;20.87], *p* < 0.001 vs. OR = 9.09, CI_95_ [4.61;18.84], *p* < 0.001, respectively).

Favourable combinations of B-mode and qualitative CEUS features in the prediction of thyroid cancer were also noted for stiffness in SE assessed as Asteria score 4 and hypoenhancement or equal-to-thyroid parenchyma enhancement, increasing the odds of malignancy 11 times (OR = 11.12, CI_95_ [3.80;47.47], *p* < 0.001), while the combination of Asteria score 4 with heterogenous enhancement increased the odds of cancer 13 times (OR= 13.00, CI_95_ [5.29;37.05], *p* < 0.001; Figure 4).

In the multivariate logistic regression, two clinically significant combinations of B-mode features and CEUS examination were noted. It was found that hypoechogenicity (mild or marked) combined with heterogenous enhancement was associated with three times higher odds of cancer (OR = 3.36, CI_95_ [1.49;7.73], *p* = 0.004). An equally favourable relationship was found for non-smooth margins (irregular or ill-defined) and hypoenhancement or equal-to-thyroid parenchyma enhancement, which resulted in four times higher odds of cancer (OR = 4.03, CI_95_ [1.67;10.18], *p* = 0.002); Figure 5.

### 3.5. Pathology Results

In the group of thyroid cancers, most of the recorded lesions were in Bethesda categories V (*n* = 55 [44.7%]) and VI (*n* = 41 [33.3%]), and there were significantly lower percentages of lesions with an intermediate risk of malignancy and benign lesions, amounting to *n* = 20 (16.3%) for category IV, *n* = 5 (4.1%) for category III, and *n* = 2 (1.6%) for category II.

In the group of benign lesions, no category-VI tumours were noted, only six lesions assessed as category V were found (9.2%), and a relatively even distribution of lesions was confirmed, amounting to category IV *n* = 24 (36.9%), category III *n* = 14 (21.5%), and category II *n* = 21 (32.3%). Therefore, thyroid cancer was detected in 100% in category VI, 90.2% in category V, 45.5% in category IV, 26.3% in category III, and 8.7% in category II.

There was a significant predominance of papillary thyroid cancer (PTC) (*n* = 93; 75.6%), followed by thirteen cases (10.6%) of medullary thyroid cancer (MTC), eight cases (6.5%) of follicular thyroid cancer (FTC), five cases (4.1%) of oncocytic carcinoma, three cases (2.4%) of differentiated high-grade thyroid carcinoma, and one case of poorly differentiated cancer (0.8%). In the group of benign lesions, most had hyperplastic nodules (*n* = 34; 52.3%) and follicular adenomas (*n* = 29; 44.6%), and two cases of thyroiditis (3.1%) were found.

## 4. Discussion

### 4.1. The Highlights of the Study

The main objective of the study was to evaluate the usefulness of CEUS examination in routine clinical practice and to develop models of combined contrast-enhancement features with those assessed by B-mode examination for the detection of thyroid cancer.

The highest value of the study results from the multiparameter approach to the evaluation of thyroid nodules in the light of new diagnostics methods. In addition, we assessed the unique combinations of both B-mode and CEUS features as predictors of thyroid cancers, which increase the risk of malignancy significantly (the maximum OR ratio for the combination of US features was 21.58).

### 4.2. B-Mode Features of the Study Group

In our study, in B-mode examination, hypoechogenicity, irregular shape, ill-defined or irregular margins, taller-than-wide shape, an Asteria score of 4 in SE, and microcalcifications were significantly associated with malignancy. The above observations are in line with numerous studies carried out to distinguish sonographic features that indicate an increased risk of malignancy of focal thyroid lesions in B-mode examination [6,7,24,25].

### 4.3. CEUS Features of the Study Group

In the CEUS study, features such as hypoenhancement, heterogeneous enhancement, and a slow wash-in phase were more common in malignant nodules, whereas high enhancement and a fast wash-in phase were more prevalent in benign nodules. In the multivariate logistic regression, the best results were achieved for a combination of hypoecho genicity (mild or marked) with heterogenous enhancement and non-smooth margins (irregular or ill-defined) with hypoenhancement or equal-to-thyroid parenchyma enhance ment. Our results advocate for a multimodal approach combining various US technologies to improve diagnostic accuracy that have been discussed and evaluated in numerous studies [23,26]. We presented the literature review of the selected studies and meta-analyses quoted in the manuscript in the form of two tables attached in the Appendix A [Table A4 and Table A5].

### 4.4. The Utility of CEUS Patterns Assessed in the Study in Line with the Literature Review—Malignant Nodules

The numerous papers written on the application of CEUS in thyreology have highlighted some trends in the optical patterns observed in thyroid cancer [22,23,27,28]. Our study shows that hypoenhancement is significantly more frequently observed in malignant lesions (*p* = 0.011). Similar observations were made by Zhang et al., whose study evaluating 157 thyroid nodules found that most malignant nodules had hypoenhancement, with a sensitivity of 84.15% and an accuracy of 75.16% [29]. Furthermore, this study showed that the dimension of the focal lesion influences the assessment of contrast enhancement, with smaller lesions significantly more likely to present low enhancement patterns. We can translate this observation to our study group, in which a significant difference in focal lesion dimension was noted between the groups, with the mean dimension of malignant lesions being significantly smaller than that of benign lesions (13.00 mm ± [9.0;20.75] vs. 21.5 mm ± [15.0;28.0]).

Another contrast feature characteristic of malignant lesions in our study group is heterogenous enhancement (*p* = 0.003). The heterogeneity of contrast enhancement is probably related to the presence of calcification, focal necrosis, and fibrosis in the lesions studied [30]. Our observations are consistent with the results of the paper written by Zhang et al., in which heterogeneous enhancement was shown to correlate with a malignant nodule, with high sensitivity and specificity (88.2% vs. 92.5%) [30]. Moreover, heterogeneous enhancement may be a good differentiating feature for lesions with intermediate cytopathological risk, as its presence increases the risk of thyroid cancer by 38.5% in this thyroid nodules group [31].

Another group of features analysed in our study is the evaluation of the rate of both wash-in and wash-out of contrast agent from the focal lesion compared to the surrounding thyroid parenchyma. It was noted that a slow wash-in was significantly more frequently observed in the group of malignant lesions (*p* = 0.018); these observations are consistent with the results in a paper by Xu et al., which analysed the contrast patterns of 432 focal thyroid lesions [32].

A study by Wang et al. on binary logistic regression reported the slow wash-in feature with heterogeneous and irregular enhancement, an unclear enhancement boundary, and no ring enhancement as significantly differentiating malignant and benign lesions (*p* = 0.001; *p* = 0.002; *p* = 0.023; *p* = 0.002; *p* = 0.012) [33]. In our study group, the fast wash-out feature was not shown to be a predictor of malignant change, whereas several studies have reported such a relationship [34,35].

### 4.5. The Utility of CEUS Patterns Assessed in the Study in Line with the Literature Review—Benign Nodules

Jiang et al., analysing contrast patterns in follicular thyroid adenomas, discussed the characteristic ‘fast-in and slow-out’ pattern resulting from the physiology of the vascular system in the tumour, and such statistically significant trends were also observed in our study in the benign-lesion group [36]. Jiang et al.’s paper also assessed the nature of contrast enhancement, differentiating centrifugal, centripetal, and diffuse types, and found no significant differences in this group of features between malignant and benign lesions.

Moving on to benign thyroid tumours, EFSUMB guidelines point to the ring-enhancement contrast feature as the strongest predictor, with high sensitivity and specificity [22]. The authors of the current paper presented observations consistent with the above data, showing that ring enhancement was significantly more common in the benign-lesion group (*p* = 0.006). Despite these statistically significant correlations, divergent views on the utility of the ring-enhancement feature as a predictor of benign lesions have been reported in several papers, including meta-analyses, confirming the lack of consensus and of standardised contrast patterns that are valid in routine clinical practice [27,28,32,37]. In our study group, the presence of the ring-enhancement feature was reported in 16.9% of benign lesions compared to 4.1% of thyroid carcinomas (*p* = 0.006), while univariate and multivariate logistic regression models did not distinguish this feature as significantly excluding the malignant nature of the lesion. Remaining in the group of benign lesions, the authors of the current study showed that the features of high enhancement and homogenous enhancement allow a significant differentiation of these lesions from thyroid carcinomas, which was also confirmed in numerous studies carried out with a larger number of patients [38].

### 4.6. The Usefulness of CEUS Patterns in Combination with B-Mode Features Evaluated in the Study Group in Line with the Literature Review

Over the last decade, scientists have presented papers showing the benefits of a combined assessment of focal thyroid lesions by both B-mode US, including elastography, and CEUS to increase the sensitivity and specificity of US in the prediction of thyroid cancer [22,39]. The study by Yu-Zhi et al. involving the evaluation of 145 focal lesions showed that the combined evaluation in high-resolution US, together with real-time elastography and CEUS, relevantly increased both sensitivity and specificity (87.3%; 91.5%), with the best accuracy (area under the curve [AUC] 0.935) of the methods [40]. The study showed that a combination of features, such as ill-defined margins, microcalcification, hypoechogenicity, an elastography score (ES) of 3 or 4, and two quantitative CEUS features, namely, the time-to-peak ratio (TTP) < 1.15 and peak ratio < 1.06, were independent predictors for malignant nodules that significantly increase the sensitivity to above 87% [40]. A study by Xu et al. evaluating a total of 432 focal lesions (258 malignant and 174 benign) also showed that assessment of a thyroid lesion based on a combination of B-mode features and CEUS patterns is more accurate than B-mode alone, and is characterized by high sensitivity, specificity, and accuracy (85.66%; 83.33%, and AUC 0.867, respectively) [32]. In addition, by way of logistic regression, features such as a slow wash-in, slow TTP, non-uniform and irregular enhancement, an unclear enhancement boundary, and no ring enhancement were identified as predictors of malignant lesions [31]. We also showed, using logistic regression models, statistically significant combinations of B-mode features, together with quantitative contrast patterns in the CEUS examination. It has been shown that a combination of features, such as hypoechogenicity with low or equal-to-parenchyma enhancement or with heterogenous enhancement, significantly increases the risk of malignancy (OR = 7.75 vs. OR = 6.6). In addition, a combination of marked hypoechogenicity with hypoenhancement or equal-to-thyroid parenchyma enhancement or with heterogeneous enhancement is a predictor of thyroid cancer in the lesion (OR = 21.58 vs. OR = 8.77). A statistically significant correlation was also found for non-smooth margins and hypoenhancement or equal-to-thyroid parenchyma enhancement (OR = 9.47 vs. OR = 9.09). The above observations related to the clinical benefit of using combined models are in line with the authors of those papers that confirmed that assessing thyroid nodules using CEUS combined with B-mode features significantly increases diagnostic accuracy [41]. Several papers have also evaluated the role of elastographic assessment, together with CEUS, in estimating the risk of malignancy of thyroid nodules and optimal qualification for fine-needle aspiration biopsy (FNAB) to minimise the number of unnecessary invasive procedures [42,43,44]. A study by Siu et al. showed that the combined evaluation of thyroid nodules in CEUS with SE increased the accuracy of CEUS by up to 95.41% compared to CEUS alone (85.32%) [42]. We also showed a favourable correlation of Asteria score-4 features in SE with heterogeneous enhancement in CEUS, indicating a ninefold greater risk of thyroid cancer. However, the limitations of elastography and the possibility of stiff patterns in benign thyroid diseases, such as thyroiditis or soft nodules in the case of follicular carcinoma, should be borne in mind [44].

It is worth noting that in the current paper, we demonstrated in multivariate logistic regression two statistically significant combinations of B-mode and CEUS features that are useful in the prediction of thyroid cancer, namely, hypoechogenicity with heterogeneous enhancement and non-smooth margins with hypoenhancement or equal-to-thyroid parenchyma enhancement. These observations are in line with data presented in a paper by Brandenstein et al., which reported the benefits of multiparametric US including CEUS, B-mode, and SWE in preoperative differential diagnosis between benign and malignant nodules, reaching a sensitivity of 95% and specificity of 75.49%, and highlighting—among other things—combinations of features such as ill-defined margins and heterogeneous enhancement in the prediction of thyroid cancer [45]. The statistically significant combinations of B-mode and CEUS features reported above may constitute a good diagnostic tool in estimating malignancy risk during US evaluation of focal thyroid lesions.

### 4.7. The CEUS-TIRADS Application

Another issue concerning the clinical utility of the CEUS relates to the possibility of using it to reduce unnecessary biopsies of thyroid nodules by constructing a TIRADS based on B-mode and CEUS to assess the risk of malignancy in thyroid nodules. In our study, it was observed retrospectively that EU-TIRADS category-5 lesions predominated in the thyroid carcinoma group (86.2%); however, there was a relatively even distribution of EU-TIRADS 5–3 lesions in the benign-lesion group, which justifies research into implementing CEUS in the EU-TIRADS US classifier to optimally qualify patients for FNAB. In addition, higher percentages of thyroid cancer in Bethesda category IV and III lesions (of 45.5% and 26.3%, respectively) were observed in the study group than in the publicly available cytopathology report, providing further evidence of the need to optimise the preoperative diagnostic pathway for patients with focal thyroid lesions, including by implementing CEUS testing at this stage [11]. Jingliang et al. constructed a CEUS TIRADS classification based on a retrospective study assessing 801 thyroid nodules. Qualitative US features of thyroid lesions were assessed including echogenicity, nodule shape and margin, echogenic foci, extrathyroidal extension, and nodule composition; CEUS features involving enhancement direction, peak intensity, and ring enhancement were similarly assessed, and the CEUS TIRADS had the highest AUC (of 0.93) of all the systems compared [46]. Further multicentre studies on the implementation of a standardised CEUS TIRADS classifier for use in routine clinical practice need to continue. The authors of the current study also plan to continue work on the development of the CEUS EU-TIRADS-PL classifier, taking into account the results of studies conducted to date on the use of CEUS for optimal qualification for FNAB, including, among others, the paper by Tinghui et al., which confirmed that the CEUS increases the adequacy of FNAB in high inadequate-risk thyroid nodules by avoiding unnecessary biopsies of non-enhancing lesions [47].

### 4.8. Artificial Intelligence in Optimizing the US Imaging—Future Prospects

Large-scale work is currently being carried out on the feasibility of implementing artificial intelligence (AI) to optimise the interpretation of ultrasound examinations and estimate the risk of malignancy of focal thyroid lesions. The up-to-date studies indicate that the use of AI improves the accuracy of ultrasound in the detection of thyroid cancer and has the effect of reducing the number of unnecessary FNABs [48,49]. In a review published by Sorrenti et al., several AI approaches were introduced, including their implementation for the classification of thyroid nodules and the early detection of cancers, including modifications to the American College of Radiology Thyroid Imaging Reporting and Data System (TIRADS) [50]. We plan to work on the application of AI methods, including machine learning, in the analysis of CEUS images, including the analysis of qualitative features of contrast enhancement.

### 4.9. Limitations of the Study

The main limitation of the current study is the significant predominance of malignant lesions in the study group, which is related to the fact that the study was carried out in a reference oncology centre; however, it is planned to continue the study in the clinic in order to collect a more representative group of benign lesions. An overrepresentation of thyroid cancer introduced an imbalanced proportion of malignant and benign nodules which could have influenced undervalued benign CEUS patterns. Furthermore, the interpretative difficulties of visual contrast patterns and their comparative assessment with the surrounding thyroid parenchyma may be due to the relatively high percentage of chronic autoimmune disease of the thyroid gland and the resulting parenchymal perfusion disorders. In addition, performing the test requires the establishment of intravenous access to the patient, specifically trained operators to ensure an adequate interpretation of contrast patterns, the presence of additional medical staff, and post-procedure observation of the patient. CEUS examination requirements mentioned above in a correlation with the cost-effectiveness of the procedure might have influenced the sparsity of examinations performed in ambulatory health care. Moreover, there are no standardized protocols to guide the performance of CEUS examinations. The main advantages of the study arise from the fact that the work was performed at a single reference centre; the CEUS examination was performed by a single investigator using the same US machine and its optimal settings; the visual patterns were assessed by two experienced US specialists; and the cyto- and histopathological examinations were performed by a pathomorphologist experienced in thyroid gland disorders. Work on the delineation of contrast patterns should be continued on a large scale to designate standardised contrast patterns of both benign and malignant focal thyroid lesions for implementing the CEUS examination within routine clinical practice.

## 5. Conclusions

This comprehensive study underscores the efficacy of a multimodal US approach in the evaluation of thyroid nodules. The integration of B-mode and CEUS parameters significantly enhances the diagnostic accuracy required to differentiate benign and malignant thyroid nodules. The findings of our study align with current medical research, which emphasises the value of combining traditional B-mode US characteristics with advanced CEUS features.

In particular, the combination of hypoechogenicity and non-smooth margins with patterns of enhancement in CEUS emerges as a robust predictor of malignancy.

These insights not only reinforce the pivotal role of multimodal US in thyroid nodule evaluation, but also contribute to a more nuanced understanding of the ultrasonographic characteristics indicative of thyroid cancers. This study therefore provides a valuable addition to the existing body of knowledge, suggesting a pathway toward more precise and individualised patient care in thyroid pathology.

## Figures and Tables

**Figure 1 cancers-16-01911-f001:**
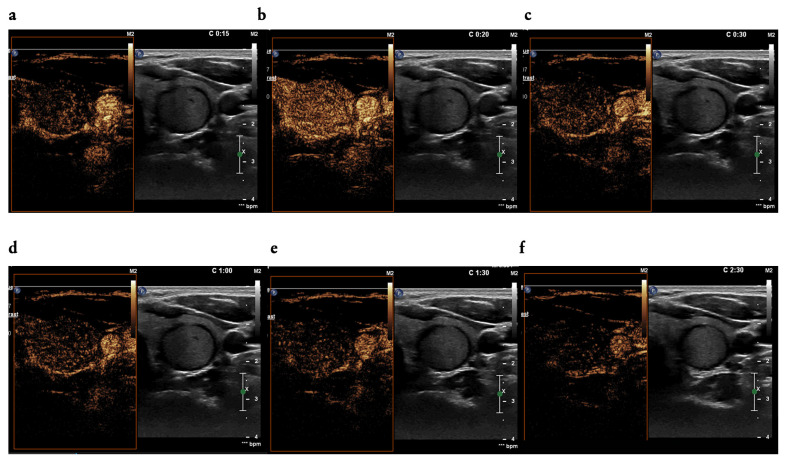
(**a**–**f**): An example of CEUS examination of a focal lesion of the left thyroid lobe: contrast-enhanced patterns after intravenous injections of the contrast agent SonoVue 1.5 mL; (**a**) wash-in phase in 15 s, (**b**) high-intensity and homogenous contrast enhancement in 20 s, (**c**) wash-out phase started in 30 s, (**d**–**f**) wash-out phase: in 60 s (**d**), in 90 s (**e**) and at the end of the examination in 150 s (**f**).

**Figure 2 cancers-16-01911-f002:**
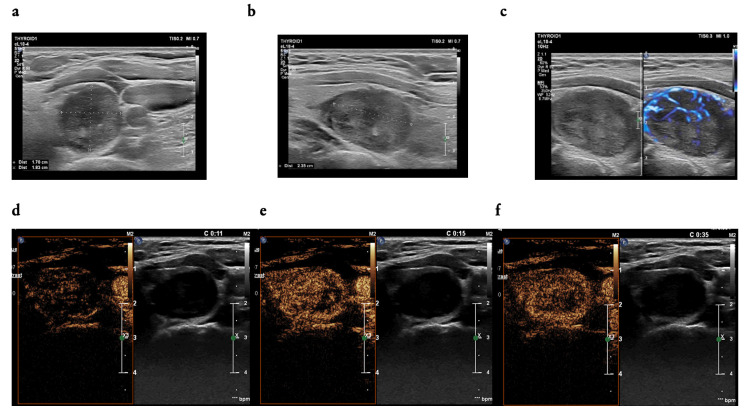
(**a**–**f**): A 42-year-old woman—a focal lesion of the left thyroid lobe; (**a**) two dimensions of the thyroid nodule (width and depth) measured in a transverse position of a transducer, (**b**) one dimension of the thyroid nodules (length) measured in a longitudinal position of a transducer; (**a**,**b**) in B-mode ultrasound examination: solid, mildly hypoechoic, heterogenous, with non-parallel orientation and (**c**) with mixed vascularity pattern in micro flow imaging (MFI); in the CEUS examination: contrast enhancement occurred in 10 s (**d**), with intensity comparable to the thyroid parenchyma and heterogeneous (**e**), with a fast wash-out phase that began within 35 s (**f**), EU-TIRADS 5. FNAB (fine needle aspiration biopsy)—Bethesda cat. IV, result of histopathological examination: follicular thyroid carcinoma (FTC).

**Figure 3 cancers-16-01911-f003:**
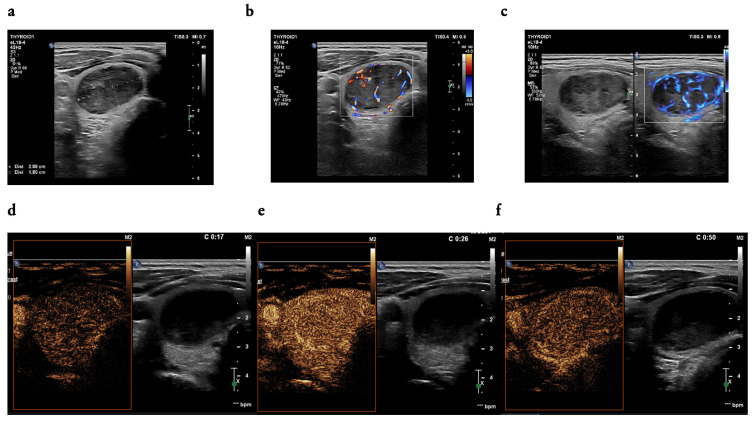
(**a**–**f**): A 46-year-old man—a focal lesion of the right thyroid lobe; (**a**) two dimensions of the thyroid nodule (width and depth) measured in a transverse position of a transducer; in B-mode ultrasound examination: solid, markedly hypoechoic, heterogenous (**a**), with mixed vascularity pattern on colour CD (colour Doppler) (**b**), and MVI (**c**); in the CEUS examination: contrast enhancement in the wash-in phase in 17 s (**d**), with high-enhancement and homogenous (**e**), with a slow wash-out phase (**f**), EU-TIRADS 5, FNAB (fine needle aspiration biopsy)—Bethesda cat. II, result of histopathological examination: hyperplastic nodule.

**Figure 4 cancers-16-01911-f004:**
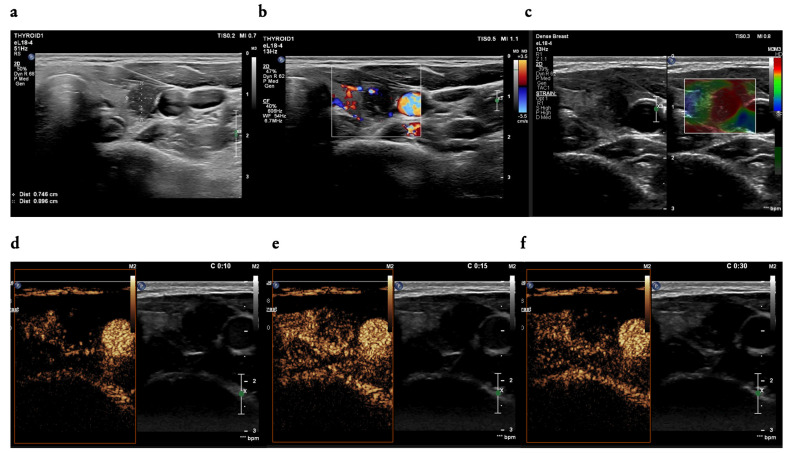
(**a**–**f**): A 38-year-old woman—a focal lesion of the left thyroid lobe; (**a**) two dimensions of the thyroid nodule (width and depth) measured in a transverse position of a transducer; in B-mode ultrasound examination: solid, mildly hypoechoic, with non-parallel orientation, ill-defined margin (**a**), and peripheral vascularity of CD (**b**), in strain elastography (SE) Asteria score 4 (**c**); in the CEUS examination: contrast enhancement occurred in 10 s (**d**), hypoenhancement and heterogeneous (**e**), with a fast wash-out phase that started within 30 s (**f**), EU-TIRADS 5, FNAB (fine-needle aspiration biopsy)—Bethesda cat. V, result of histopathological examination: papillary thyroid carcinoma (PTC).

**Figure 5 cancers-16-01911-f005:**
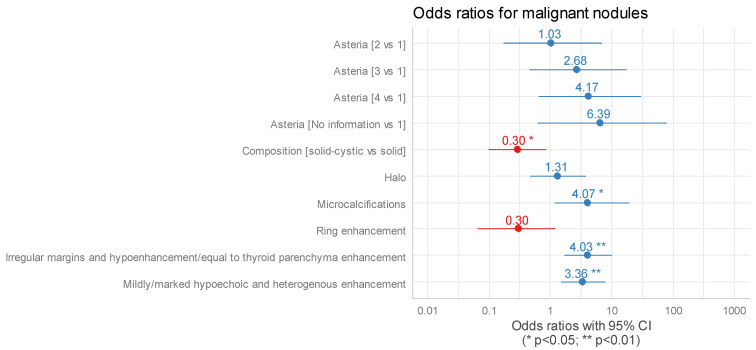
Outcomes of multivariate logistic regression for malignant nodules (blue lines with numbers—confidence interval for OR above 1.00, red lines with numbers—confidence interval for OR below 1.00). OR—odds ratio [Table 4].

**Table 1 cancers-16-01911-t001:** Study population and ultrasound (US) characteristics of the thyroid nodules. * IQR—interquartile range. SE—strain elastography.

Variable	Malignant (*n* = 123, 65.4%)	Benign (*n* = 65, 34.6%)	*p*-Value
Size, mm, median (IQR) *	13.00 (9.00;20.75)	21.50 (15.00;28.00)	<0.001
Echogenicity			
Markedly hypoechoic	67 (54.5)	11 (16.9)	<0.001
Hyperechoic	0 (0.0)	3 (4.6)	0.040
Isoechoic	6 (4.9)	23 (35.4)	<0.001
Composition			
Solid	115 (93.5)	48 (73.8)	<0.001
Solid-cystic	8 (6.5)	17 (26.2)	
Shape			
Oval	94 (76.4)	63 (96.9)	0.001
Irregular	28 (22.8)	1 (1.5)	<0.001
Margins			
Smooth	19 (15.4)	43 (66.2)	<0.001
Ill-defined	52 (42.3)	13 (20.0)	0.004
Irregular	52 (42.3)	9 (13.8)	<0.001
Margins irregular–angular	19 (15.4)	2 (3.1)	0.020
Margins irregular–spicular	12 (9.8)	0 (0.0)	0.009
Halo/rim	13 (10.6)	20 (30.8)	0.001
Halo by type			
Thin	7 (5.7)	14 (21.5)	0.002
Microcalcification	40 (32.5)	3 (4.6)	<0.001
Asteria score in SE			
2	18 (14.6)	29 (44.6)	<0.001
4	47 (38.2)	8 (12.3)	<0.001
EU-TIRADS category			
3	6 (4.9)	20 (30.8)	<0.001
4	11 (8.9)	22 (33.8)	<0.001
5	106 (86.2)	23 (35.4)	<0.001
Bethesda category			
II	2 (1.6)	21 (32.3)	<0.001
III	5 (4.1)	14 (21.5)	<0.001
IV	20 (16.3)	24 (36.9)	0.003
V	55 (44.7)	6 (9.2)	<0.001
VI	41 (33.3)	0 (0.0)	<0.001

**Table 2 cancers-16-01911-t002:** Logistic regression outcomes for determining the risk of malignant nodules according to ultrasound (US) features. Outcomes of Asteria 2 and Asteria 3 in strain elastography (SE) are not included in the table and can be viewed in Table A2 in the Appendix A. SE—strain elastography.

B-Mode	CEUS	Univariate Model			Multivariate Model		
		OR	95% CI	*p*-Value	OR	95% CI	*p*-Value
Size, mm	–	0.96	0.93–0.98	0.002	–	–	–
Markedly hypoechoic (vs. mildly hypoechoic)	–	3.41	1.59–7.76	0.002	–	–	–
Isoechoic (vs. mildly/markedly hypoechoic)	–	0.15	0.05–0.38	<0.001	–	–	–
Hypoechoic (mildly/markedly)	–	13.00	5.29–37.05	<0.001	–	–	–
Composition, solid-cystic (vs. solid)	–	0.20	0.08–0.47	<0.001	0.30	0.10–0.85	0.027
Shape, oval (vs. irregular)	–	0.05	0.00–0.26	0.004	–	–	–
Margins, ill-defined (vs. smooth)	–	9.05	4.12–21.08	<0.001	–	–	–
Margins, irregular (vs. smooth)	–	13.08	5.58–33.50	<0.001	–	–	–
Margins irregular–angular	–	5.75	1.60–36.89	0.021	–	–	–
Halo (vs. no halo)	–	0.27	0.12–0.57	<0.001	1.31	0.46–3.80	0.609
Microcalcifications	–	9.96	3.41–42.48	<0.001	4.07	1.18–19.21	0.042
Vascularity type II (vs. mixed)	–	3.39	1.65–7.46	0.001	–	–	–
Asteria score 4 in SE (vs. Asteria 1)	–	7.34	1.63–35.92	0.010	4.17	0.64–29.95	0.141

**Table 3 cancers-16-01911-t003:** Logistic regression outcomes for determining the risk of malignant nodules according to contrast-enhanced ultrasound (CEUS) examination.

		Univariate Model			Multivariate Model		
CEUS Contrast Enhancement Features		OR	95% CI	*p*	OR	95% CI	*p*
High enhancement (vs. lower than thyroid parenchyma)	–	0.23	0.10–0.50	<0.001	–	–	–
Heterogenous (vs. homogenous)	–	2.87	1.46–5.71	0.002	–	–	–
Ring enhancement	–	0.21	0.06–0.60	0.005	0.30	0.07–1.19	0.098
Fast wash-out phase (vs. no wash-out)	–	0.16	0.02–0.60	0.018	–	–	–
Equal to thyroid parenchyma wash-out phase (vs. no wash-out)	–	0.17	0.03–0.66	0.025	–	–	–
Slow wash-out phase (vs. no wash-out)	–	0.05	0.01–0.22	<0.001	–	–	–

**Table 4 cancers-16-01911-t004:** Logistic regression outcomes for determining the risk of malignant nodules. SE—strain elastography. CEUS—contrast-enhanced ultrasound.

B-Mode Features	CEUS Features	Univariate Model			Multivariate Model		
		OR	95% CI	*p*-Value	OR	95% CI	*p*-Value
Hypoechogenicity (mild/marked)	Hypoenhancement/equal-to-thyroid parenchyma enhancement	7.75	3.95–15.93	<0.001	–	–	–
Hypoechogenicity (mild/marked)	Heterogenous enhancement	6.60	3.42–13.10	<0.001	3.36	1.49–7.73	0.004
Marked hypoechogenicity	Hypoenhancement/equal-to-thyroid parenchyma enhancement	21.58	6.34–135.23	<0.001	–	–	–
Marked hypoechogenicity	Heterogenous enhancement	8.77	3.78–24.07	<0.001	–	–	–
Non-smooth margins (ill-defined/irregular)	Hypoenhancement/equal-to-thyroid parenchyma enhancement	9.47	4.63–20.87	<0.001	4.03	1.67–10.18	0.002
Non-smooth margins (ill-defined/irregular)	Heterogenous enhancement	9.09	4.61–18.84	<0.001	–	–	–
Asteria score 4 in SE	Hypoenhancement/equal-to-thyroid parenchyma enhancement	11.12	3.80–47.47	<0.001	–	–	–
Asteria score 4 in SE	Heterogenous enhancement	13.00	5.29–37.05	<0.001	–	–	–

**Table 5 cancers-16-01911-t005:** Characteristics of the thyroid lesions according to contrast-enhanced ultrasound (CEUS) examination.

Variable Contrast Enhancement Features	Malignant (*n* = 123, 65.4%)	Benign (*n* = 65, 34.6%)	*p*-Value
Intensity			
High enhancement	33 (26.8)	36 (55.4)	<0.001
Lower than thyroid parenchyma	44 (35.8)	11 (16.9)	0.011
Uniformity			
Homogenous	22 (17.9)	25 (38.5)	0.003
Heterogenous	101 (82.1)	40 (61.5)	
Wash-in phase			
Slow	40 (32.5)	10 (15.4)	0.018
Fast	29 (23.6)	26 (40.0)	0.029
Ring enhancement	5 (4.1)	11 (16.9)	0.006
Wash-out phase			
Slow	12 (9.8)	19 (29.2)	0.001

## Data Availability

Data are contained within the article.

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
