# Peer review of "The Utility of Contrast-Enhanced Ultrasound (CEUS) in Assessing the Risk of Malignancy in Thyroid Nodules"

_cancers, 2024, doi:10.3390/cancers16101911_

Round 1
Reviewer 1 Report
Comments and Suggestions for Authors
Authors wanted to underlie the role of CEUS combined to US findings of thyroid nodules.
The US score systems (both EU-TIRADS and ACR-TIRADS) are able to define a suspicious nodule rather than a benign one, and this is he criteria used to perform fine needle aspiration (FNA).
The results of this paper show an addictive effect of CEUS, but, even if not so expensive, it adds another step in the evaluation of nodules taking more time and nursing staff not always presenti during FNA.
Comments on the Quality of English LanguageThe paper is easy to read
Author Response
Reviewer 1
We would like to thank the Reviewer for valuable opinion and suggestions. It is our honour to have opportunity to perform our result. The positive Reviewer’s opinion will be motivation for further research in thyroid ultrasound field.

Reviewer 2 Report
Comments and Suggestions for Authors
Thanks for the effort of this interesting study. Here are my comments below.
1. Sensitivity and specificity of diagnostic modality can be enhanced by machine learning. It should be introduced in introduction part. The paper below should be included in the manuscript.
Mao, Y.-J.; Zha, L.-W.; Tam, A.Y.-C.; Lim, H.-J.; Cheung, A.K.-Y.; Zhang, Y.-Q.; Ni, M.; Cheung, J.C.-W.; Wong, D.W.-C. Endocrine Tumor Classification via Machine-Learning-Based Elastography: A Systematic Scoping Review. Cancers 2023, 15, 837.
2. US frequency can significantly affect the measurement of length, depth and width of the lesion. What was the exact frequency used in the examination?
3. An illusion or figures of US Image for measurement should be provided for how these dimensions measured.
4. Interobserver variation should be investigated, although they have more 10 years experience.
5. Since there are many tables and items. It should have a summary table to show all items with statistical significant difference. These could be saline feature for further analysis or facilitating machine learning model development in future study.
6. The measurement lines in the figure 1,2 and 3 are not clear. It may paint with different colour or increased thickness.
7. There should have image or figure to illustrate the features in each table.
8. Please highlight the key findings at the beginning of discussion.
9. There are too many comparison among different studies. It is very hard to follow. Please summary in form of table and consider to move some of these reviewed studies to introduction or literature review.
10. The organization of discussion must be conducted. Multimodal US results should be highlighted.
11. Since there are many salient features found, future work should include using machine learning to improve the accuracy.
Author Response
Reviewer 2
We would like to thank the Reviewer for valuable suggestions and feedback. The reviewer’s comments were very insightful and have helped us to improve the quality of our manuscript. In accordance with the reviewer’s comments, revisions to the text have been marked using the highlighting feature. Our changes are typed in yellow.
Changes introduce in the article:
- We added in the Introduction part information about machine learning:
,Nowdays, machine learning models are used in order to improve sensitivity, specificity and accuracy of diagnostic modalities, including the reliability of elastography on the classification of thyroid nodules’, we attached the reference as below:
Mao, Y.J.; Zha, L.W.; Tam, A.Y.; Lim, H.J.; Cheung, A.K.; Zhang, Y.Q.; Ni, M.; Cheung, J.C.; Wong, D.W. Endocrine Tumor Classification via Machine-Learning-Based Elastography: A Systematic Scoping Review. Cancers (Basel). 2023, 15, 837.
- We specified the frequency of linear probe in the corrected sentence as below:
‘All 188 patients underwent US examination on a high-performance US device (Philips, EpiQ 5) using a linear probe with a frequency of 18.0 MHz’.
- We specified in the manuscript and in description of figures how dimensions of thyroid nodules were measured: ‘length (in longitudinal position of a transducer), depth, and width (in transverse position of a transducer)’.
- Firstly, all US images and DICOM memory loops were evaluated by specialists separately. On the second step, all database were discussed together and interobserver disagreement were assessed in approximately ten per cent of cases, in which consensus was reached decisively.
- We attached three summary tables (Table 6,7,8) in the appendix, which we did not performed in the main manuscript part. The Table 8 was introduced in revised version so as to perform sensitivity, specificity, accuracy, PPV, NPV of the dominant B-mode and CEUS features.
- We corrected the visibility of the measurements lines in attached figures.
- In presented figures B-mode images (including echogenicity, margins, shape, vascularization type in color Doppler and micro flow images, strain elastography) and CEUS patterns were presented, according to ultrasound features presented in tables.
- We highlighted the key findings of the manuscript at the beginning of discussion:
‘The highest value of the study results from the multiparameter approach to the evaluation thyroid nodules in the light of new diagnostics methods. In addition, we assessed the uniqe combinations of both B-mode and CEUS features as predictors of thyroid cancers, which increase the risk of malignancy significantly (the maximum OR ratios for the combination of US features ranged 21.58).
- We constructed two tables with selection of quoted in the manuscript studies, including meta-analyses. We attached this tables in the appendix in order to make the study more readable.
- We reorganized a construction of the discussion. At the beginning we highlighted the multiparameter US approach. We removed the part of the manuscript described in details results of Remonti et al. study, instead of that we marked the importance of multimodality of our study, and referred to tables with literature reviews in the appendix. Moreover we improved readability of the discussion part by adding more subtitles.
- We introduced additional part of the discussion describing the possibilities of using artificial intelligence in ultrasound imaging, involving machine learning.
We hope that the revisions to the manuscript and accompanying responses will be sufficient to make our manuscript suitable for publication in the Journal – Cancers.
Reviewer 3 Report
Comments and Suggestions for Authors
Manuscript title: The utility of contrast-enhanced ultrasound (CEUS) in assessing the risk of malignancy in thyroid nodules.
Reviewer comments:
The study by Żyłka et al. evaluates the role of contrast-enhanced ultrasound (CEUS) as a means to assess the risk of malignancy in thyroid nodules. To this end, the authors conducted a clinical study enrolling 188 patients with focal thyroid lesions in Bethesda categories II-VI who were candidates for surgical treatment and performed a B-mode ultrasound (US) followed by a CEUS of target lesions. Then, they developed univariate and multivariate logistic regression models to identify US and CEUS features correlating with the risk of malignancy detected in the postoperative histopathological examination. In multivariate logistic regression, the authors identified two combinations of B-mode and CEUS features associated with a significantly increased probability of thyroid cancer, namely non-smooth margins with hypoenhancement or equal to thyroid parenchyma enhancement, and hypoechogenicity with heterogeneous enhancement.
The problem addressed by the authors is particularly relevant for the field, given the lack of an unequivocal sonographic pattern indicative of thyroid cancer, with the subsequent need of reliable methods to risk-stratify thyroid lesions and avoid unnecessary surgeries, especially in the case of cytologically indeterminate nodules. As the authors point out, similar works evaluating the use of combinations of B-mode, elastography, and CEUS findings to aid in the differential diagnosis of thyroid lesions have already been published, thus undermining the novelty of the current study. Nevertheless, the findings reported are still compelling and merit consideration.
Overall, the study design and methodology employed are sound, and the conclusions are in accordance with the observed findings. The manuscript is clear and well-presented, with appropriate figures and easy-to-interpret tables, and the cited references are relevant. However, there are some major and minor issues that need to be addressed.
Major comments:
-
The development of noninvasive methods to achieve a more accurate characterization of thyroid nodules is an area of extensive research efforts, since this could avoid unnecessary surgical procedures and fine-needle aspirations (FNAs). As highlighted in this and in other studies, B-mode US and CEUS hold promise in this context, because they could allow to develop risk stratification systems to identify patients with a high likelihood of having benign lesions and prevent them from undergoing invasive procedures. The patients enrolled in this study were all candidates for thyroid surgery, but their nodules belonged to different categories of the Bethesda classification (i.e., from category II [benign] to category VI [malignant]). Could the authors perform a subgroup analysis to assess the correlation between CEUS patterns that were found to be more dominant in subjects with histopathologically benign lesions (e.g., ring enhancement, fast wash-in phase) and the Bethesda class assigned to the nodules demonstrating such imaging features? Such an analysis would be of great interest, since it could allow to identify imaging characteristics that are indicative of a low probability of malignancy, thus potentially preventing patients with benign lesions from undergoing unnecessary FNA examinations;
-
The authors built binary logistic regression models and expressed the association of US findings with the odds of a histopathological diagnosis of thyroid cancer using odds ratios (ORs). However, the diagnostic accuracy, sensitivity and specificity for the detection of malignancy of the single US findings are missing;
-
As reported in the “Discussion” section, a key limitation of this study is that it was conducted in a reference oncology center, thus yielding an overrepresentation of malignant thyroid lesions. The authors should elaborate further on the possible implications of this pitfall, highlighting the ways in which it could have influenced the significance of their results;
-
Other limitations should be discussed, including the fact that CEUS requires specifically trained operators to ensure an adequate interpretation of contrast patterns and that there are no standardized protocols to guide the performance of CEUS examinations;
-
As pointed out in the manuscript, CEUS entails the intravenous administration of contrast medium and the presence of additional medical staff. Could the authors include a brief discussion regarding the way in which these requirements could hinder the applicability of CEUS to routine clinical practice? Also, a comment on the cost-effectiveness of CEUS should be added.
Minor comments:
-
Line 51: “malignant cancer” is redundant. Please remove the word “malignant”;
-
Line 66: the part that reads “a contrast agent constituting microvesicles containing sulphur hexafluoride” is not clear and should be rephrased;
-
Line 159: what does the “1” superscript mean in “0.251”?;
-
Line 175 should read “The median dimensions were…” instead of “The mean median dimensions were”;
-
Line 187: please replace “The table below” with a more specific notation, such as “Table 1”, since the manuscript contains multiple tables;
-
Lines 211-212 are not clear and should be rephrased (e.g., “All patients qualified for CEUS examination, which revealed that qualitative features of contrast enhancement differed significantly between groups.”);
-
Line 224: “contrast enhancement features” should be plural;
-
Line 234: “the” is unnecessary;
-
Line 235: “increased” should be in the past tense;
-
Lines 241, 245, 250, 254, 256: when reporting ORs, it is advisable to describe the results in terms of “increased/reduced odds” rather than “increased/reduced risk” (e.g., lines 240-242: “the hypoechogenicity of the focal lesion (mild or marked hypoechogenicity) analysed with heterogeneous contrast enhancement increased the odds of malignancy of the focal lesion by almost seven times (OR=6.6, CI95 [3.42;13.10], p<0.001)”);
-
Line 264: please add the reference to Table 5 in addition to Figure 4;
-
Line 373: “showed” should be in the past tense;
-
Lines 377-382 (from “A study by…” to “respectively”) are unclear and should be rephrased;
-
Lines 408-412 repeat the same findings reported in lines 387-394;
-
In general, it would be advisable for the authors to refer to themselves as “we” rather than “the authors of the current paper” (e.g., in lines 353, 408, etc.);
-
In the “References” section, the same article is cited twice (references 5 and 24).
Additional comments: Ethics Statement and Data Availability Statement are missing.
Author Response
Reviewer 3
We would like to thank the Reviewer for valuable suggestions. The reviewer’s remarks were very insightful and have helped us to improve the quality of our manuscript.
In accordance with the reviewer’s comments, revisions to the text have been marked using the highlighting yellow feature. It is our honour to have opportunity to respone for the review and we hope that the manuscript and accompanying responses will be appropriate for further publication process.
- We performed a subgroup analysis to assess the correlation between CEUS patterns that might be found to be more dominant in subjects with histopathologically benign lesions and with correlation with Bethesda class. Unfortunately we did not revealed statistical significant correlation in order to differentiate nodules in this group. We attached below the table with the description, but we did not introduced the results in the manuscript, because of insignificant statistical findings. We presented dominant benign CEUS features in separate tables, but without correlation in Bethesda class. Thank you for the insightful suggestion, we will continue this observation in further research.
|
Variable |
Bethesda category of benign nodules (n = 65) |
p |
|||
|
II (n = 21, 32.3%) |
III (n = 14, 21.5%) |
IV (n = 24, 36.9%) |
V (n = 6, 9.2%) |
||
|
Intensity - high enhancement |
12 (57.1) |
9 (64.3) |
13 (54.2) |
2 (33.3) |
0.680 |
|
Wash-in phase – fast |
7 (33.3) |
7 (50.0) |
9 (37.5) |
3 (50.0) |
0.749 |
|
Uniformity – homogenous |
7 (33.3) |
6 (42.9) |
11 (45.8) |
1 (16.7) |
0.596 |
|
Ring enhancement |
3 (14.3) |
4 (28.6) |
2 (8.3) |
2 (33.3) |
0.243 |
|
Wash-out phase – slow |
5 (23.8) |
5 (35.7) |
7 (29.2) |
2 (33.3) |
0.829 |
Table. Dependency of selected features on Bethesda category, subgroup with benign nodules
2.We performed a statistical analysis referred to the single US findings with the diagnostic accuracy, sensitivity, specificity for the detection of malignancy of the single US findings.
We attached the table in the appendix, in order to make the manuscript more legible (Table 8 - ROC for predicting malignant nodules, sorted by AUC; UC – area under curve, CI – confidence interval, PPV – positive predictive value, NPV – negative predictive value).
Thank you for that remarks that we attached in the manuscript in limitation parts:
‘An overrepresentation of thyroid cancer introduced unbalanced proportion between malignant and benign nodules which could influenced undervaluated benign CEUS patterns’.
- and 5.
Thank you for the valuable remarks, which we describe in limitation part in revised manuscript:
‘In addition, performing the test requires the establishment of intravenous access to the patient, specifically trained operators to ensure an adequate interpretation of contrast patterns, the presence of additional medical staff, and post-procedure observation of the patient. CEUS examinations requirements mentioned above in a correlation with the cosf-effectiveness of the procedure might influenced the sparsity of examinations performed in an ambulatory health care. Moreover, there are no standardized protocols to guide the performance of CEUS examinations’.
Thank you very much for the precise revision and remarks. We introduced correction in the manuscript.
Minor comments:
Line 51 – ‘malignant’ was removed.
Line 66 – the correction of the sentences was done ‘contrast agent composed of microvesicles containing sulphur hexafluoride surrounded by phospholipids and palmitic acid’.
Line 159 – superscript ‘1’ was removed
Line 175 – the word ‘mean’ was removed .
Line 187 – the sentences and quotation was corrected: ‘Moreover, the data presented in the table 1 show the features that differentiated relevantly the nature of the focal lesions’.
Lines 211-212 – we introduced corrections of the sentences:
‘All patients were qualified for CEUS examination. Qualitative features of contrast enhancement evaluated in the study differed significantly between malignant and benign groups.’
Line 224 – plural form was introduced.
Line 234 – ‘the’ was removed.
Line 235 – ‘increased’ was written in the past tense.
Lines 241, 242, 250, 254, 256 – corrections with terms ‘odds’ were performed.
Line 264 – we added the reference Table 5 to the figure 4.
Line 373 – we made correction – ‘showed’ in the past tense.
Line 377-382 – we changed the construction of the sentence as below:
‘A study by Xu et al. evaluating a total of 432 focal lesions (258 malignant and 174 benign) also showed that assessment of a focal thyroid lesion based on a combination of B-mode features and CEUS patterns is more accurate than B-mode alone and characterized with high sensitivity, specificity, and accuracy (85.66%; 83.33%, and AUC 0.867, respectively)’.
Line 408-412 we quoted only reference 40, instead of both 40 and 31 - quoted in previos lines.
Line 353, 408, we change the form on ‘We’.
Reference section was corrected and the same quotations 5 and 24 was removed.
Reviewer 4 Report
Comments and Suggestions for Authors
The article is well written. The following is my suggestion:
- 1. Strengthen the introduction by referencing additional CEUS-related articles, such as those found at https://pubmed.ncbi.nlm.nih.gov/22173085/ and https://pubmed.ncbi.nlm.nih.gov/38537180/.
- 2. Include the study's hypothesis for clarity.
- 3. Improve readability by adding more subtitles.
- 4. Verify intra-rater and inter-rater reliability of ultrasound image interpretation before data collection.
- 5. Enhance comprehension by incorporating a flow diagram illustrating the study's progression.
Author Response
Reviewer 4
We would like to thank the Reviewer for valuable suggestions. The reviewer’s remarks were very insightful and have helped us to improve the quality of our manuscript.
In accordance with the reviewer’s comments, revisions to the text have been marked using the highlighting yellow feature. It is our honour to have opportunity to respone for the review and we hope that the manuscript and accompanying responses will be appropriate for further publication process.
- Thank you for the insight remarks presenting CEUS method as valuable and promising tool in different medical field. We strengthened the introduction part by referring to studies presenting wide range of indication of the CEUS examination.
‘The CEUS method is in clinical use in different medical field, involved imaging the liver lesions and wide range non-hepatic indications referred to musculoskeletal medicine or assessment urinary tract pathology’.
Chen, S.Y.; Wang, Y.W.; Chen, W.S.; Hsiao, M.Y Update of Contrast-enhanced Ultrasound in Musculoskeletal Medicine: Clinical Perspectives - A Review. J Med Ultrasound. 2023, 31, 92-100.
Dietrich, C.F.; Nolsøe, C.P.; Barr, R.G.; Berzigotti, A.; Burns, P.N.; Cantisani, V.; Chammas, M.C.; Chaubal, N.; Choi, B.I.; Clevert, D.A.; Cui, X.; Dong, Y.; D'Onofrio, M.; Fowlkes, J.B.; Gilja, O.H.; Huang, P.; Ignee, A.; Jenssen, C.; Kono, Y.; Kudo, M.; Lassau, N.; Lee, W.J.; Lee, J.Y.; Liang, P.; Lim, A.; Lyshchik, A.; Meloni, M.F.; Correas, J.M.; Minami, Y.; Moriyasu, F.; Nicolau, C.; Piscaglia, .F; Saftoiu, A.; Sidhu, P.S.; Sporea, I.; Torzilli, G.; Xie, X.; Zheng, R. Guidelines and Good Clinical Practice Recommendations for Contrast-Enhanced Ultrasound (CEUS) in the Liver-Update 2020 WFUMB in Cooperation with EFSUMB, AFSUMB, AIUM, and FLAUS. Ultrasound Med Biol. 2020, 46, 2579-2604.
- Thank you for the suggestions reffered to hypothesis of the study.
We presented the main objective in Introduction and Discussion part.
Introduction
‘The main objective of this paper is to evaluate the clinical utility of the application of CEUS’s examination features in combination with the analysis of B-mode features, including elastography, in the differential diagnosis of focal thyroid lesions’.
Discussion
‘The main objective of the study was to evaluate the usefulness of CEUS examination in routine clinical practice and to develop models of combined contrast enhancement features with those assessed by B-mode examination for the detection of thyroid cancer’.
We added in Discussion the sentence below:
‘The highest value of the study results from the multiparameter approach to the evaluation thyroid nodules in the light of new diagnostics methods. In addition, we assessed the unique combinations of both B-mode and CEUS features as predictors of thyroid cancers, which increase the risk of malignancy significantly’.
- Thank you for the valuable suggestion. W added more subtitles in Discussion in order to improve readability.
- We presented our disagreements in the sentences attached to the manuscript:
‘Firstly, all US images and DICOM memory loops were evaluated by specialists separately. On the second step, all database were discussed together and interobserver disagreement were assessed in approximately ten per cent of cases, in which consensus was reached decisively’. - We attached figures visualised the enhance comprehension and illustration the study’s progression (Figure 1).
Round 2
Reviewer 2 Report
Comments and Suggestions for Authors
The questions have been well addressed. The format of the tables should be improved. You may consider using an Excel cell-embedded colour bar to highlight the difference among those values.
Author Response
We would like to thank the Reviewer for valuable suggestions and feedback. The reviewer’s comments were very insightful and have helped us to improve the quality of our manuscript. We would like to thank the Reviewer for the consideration and courtesy. In accordance with the reviewer’s comments, we introduced graphic changes in tables and we highlighted every other line typed in gray in order to mark difference among different values. Moreover, the word count of the Simple Summary section was extended to 148 words. Futhermore, we will follow all instructions of Academic Editor in order to improve all parts of manuscript including tables and figures. It is our honour to have opportunity to perform our result. The positive reviewer’s opinion will be motivation for further research in thyroid ultrasound fields.
We hope that the revisions to the manuscript and accompanying responses will be sufficient to make our manuscript suitable for publication in the Journal – Cancers.
Reviewer 3 Report
Comments and Suggestions for Authors
n/a
Author Response
We would like to thank the Reviewer for valuable suggestions and feedback. The reviewer’s comments were very insightful and have helped us to improve the quality of our manuscript. We would like to thank the Reviewer for the consideration and courtesy. We introduced graphic changes in tables and we highlighted every other line typed in gray in order to mark difference among different values. Moreover, the word count of the Simple Summary section was extended to 148 words. Futhermore, we will follow all instructions of Academic Editor in order to improve all parts of manuscript including tables and figures. It is our honour to have opportunity to perform our result. The positive reviewer’s opinion will be motivation for further research in thyroid ultrasound fields.
We hope that the revisions to the manuscript and accompanying responses will be sufficient to make our manuscript suitable for publication in the Journal – Cancers.